# Fixing Weight Decay Regularization in Adam

## Abstract

We note that common implementations of adaptive gradient algorithms, such as Adam, limit the potential benefit of weight decay regularization, because the weights do not decay multiplicatively (as would be expected for standard weight decay) but by an additive constant factor. We propose a simple way to resolve this issue by decoupling weight decay and the optimization steps taken w.r.t. the loss function. We provide empirical evidence that our proposed modification (i) decouples the optimal choice of weight decay factor from the setting of the learning rate for both standard SGD and Adam, and (ii) substantially improves Adam's generalization performance, allowing it to compete with SGD with momentum on image classification datasets (on which it was previously typically outperformed by the latter). We also demonstrate that longer optimization runs require smaller weight decay values for optimal results and introduce a normalized variant of weight decay to reduce this dependence. Finally, we propose a version of Adam with warm restarts (AdamWR) that has strong anytime performance while achieving state-of-the-art results on CIFAR-10 and ImageNet32x32. Our source code will become available after the review process.

## 1 Introduction

Adaptive gradient methods, such as AdaGrad (Duchi et al., 2011), RMSProp (Tieleman & Hinton, 2012), and Adam (Kingma & Ba, 2014) have become a default method of choice for training feed-forward and recurrent neural networks (Xu et al., 2015; Gregor et al., 2015; Radford et al., 2015). Nevertheless, state-of-the-art results for popular image classification datasets, such as CIFAR-10 and CIFAR-100 Krizhevsky (2009), are still obtained by applying SGD with momentum (Huang et al., 2016; 2017; Loshchilov & Hutter, 2016; Gastaldi, 2017). Furthermore, Wilson et al. (2017) suggested that adaptive gradient methods do not generalize as well as SGD with momentum when tested on a diverse set of deep learning tasks such as image classification, character-level language modeling and constituency parsing. Different hypotheses about the origins of this worse generalization have been investigated, such as the presence of sharp local minima (Keskar et al., 2016; Dinh et al., 2017) and inherent problems of adaptive gradient methods (Wilson et al., 2017). In this paper, we show that a major factor in the poor generalization of the most popular adaptive gradient method, Adam, lies in its dysfunctional implementation of weight decay; the issue we identify in Adam also pertains to other adaptive gradient methods.

Specifically, our analysis of Adam given in this paper leads to the following observations:

**The standard way to implement $L_2$ regularization/weight decay in Adam is dysfunctional.**
One possible explanation why Adam and other adaptive gradient methods might be outperformed by SGD with momentum is that $L_2$ regularization/weight decay are implemented suboptimally in common deep learning libraries. Therefore, on tasks/datasets where the use of $L_2$ regularization is beneficial (e.g., on many popular image classification datasets), Adam leads to worse results than SGD with momentum (for which $L_2$ regularization behaves as expected).

**$L_2$ regularization and weight decay are not the same thing.** Contrary to common belief, the two techniques are not equivalent. For SGD, they can be made equivalent by a reparameterization of the weight decay factor based on the learning rate; this is not the case for Adam. In particular, when combined with adaptive gradients, $L_2$ regularization leads to weights with large gradients being regularized less than they would be when using weight decay.

**Optimal weight decay is a function (among other things) of the total number of batch passes/weight updates.**
Our empirical analysis of Adam suggests that the longer the runtime/number of batch passes to be performed, the smaller the optimal weight decay. This effect tends to be neglected because hyperparameters are often tuned for a fixed or a comparable number of training epochs. As a result, the values of the weight decay found to perform best for short runs do not generalize to much longer runs.

Our contributions are aimed at fixing the issues described above:

**Decoupling weight decay from the gradient-based update (Section 2).** We suggest to decouple the gradient-based update from weight decay for both SGD and Adam. The resulting SGD version SGDW decouples optimal settings of the learning rate and the weight decay factor, and the resulting Adam version AdamW generalizes substantially better than Adam.

**Normalizing the values of weight decay (Section 3).** We propose to parameterize the weight decay factor as a function of the total number of batch passes. This leads to a greater invariance of the hyperparameter settings in the sense that the values found to perform best for short runs also perform well for many times longer runs.

**Adam with warm restarts and normalized weight decay (Section 4).** After we fix the weight decay in Adam and design AdamW, we introduce AdamWR to obtain strong anytime performance by performing warm restarts.

The main motivation of this paper is to fix the weight decay in Adam to make it competitive w.r.t. SGD with momentum even for those problems where it did not use to be competitive. We hope that as a result, practitioners do not need to switch between Adam and SGD anymore, which in turn should help to reduce the common issue of selecting dataset/task-specific training algorithms and their hyperparameters.

## 2    DECOUPLING THE WEIGHT DECAY FROM THE GRADIENT-BASED UPDATE

In the weight decay described by Hanson & Pratt (1988), the weights $\boldsymbol{x}$ decay exponentially as

$$\boldsymbol{x}_{t+1} = (1 - w_t)\boldsymbol{x}_t - \alpha_t \nabla f_t(\boldsymbol{x}_t), \tag{1}$$

where $w_t$ defines the rate of the weight decay at time-step $t$ and $\nabla f_t(\boldsymbol{x}_t)$ is the $t$-th batch gradient multiplied by a learning rate $\alpha_t$. Following Hanson & Pratt (1988), one can also modify the original batch loss $f_t(\boldsymbol{x}_t)$ and consider a bias term (also referred to as the regularization term) accounting for "costs" on weights which are, e.g., quadratic in the weight values as for $L_2$ regularization:

$$f_{t,reg}(\boldsymbol{x}_t) = f_t(\boldsymbol{x}_t) + \frac{w_t}{2} \|\boldsymbol{x}_t\|_2^2, \tag{2}$$

where $w_t$ defines the impact of the $L_2$ regularization. In order to consider the weight decay regularization, one can reformulate the objective function as in Eq. (2) or directly adjust $\nabla f_t(\boldsymbol{x}_t)$ as

$$\nabla f_{t,reg}(\boldsymbol{x}_t) = \nabla f_t(\boldsymbol{x}_t) + w_t \boldsymbol{x}_t. \tag{3}$$

Historically, stochastic gradient descent methods inherited this way of implementing the weight decay regularization.

The currently most common way (e.g., in popular libraries such as TensorFlow, Keras, PyTorch, Torch, and Lasagne) to introduce the weight decay regularization is to use the $L_2$ regularization term as in Eq. (2) or, often equivalently, to directly modify the gradient as in Eq. (3). Let's first consider the simple case of SGD with momentum; Algorithm 1 demonstrates modifying the gradients directly in this method (see line 6). The weight decay term $w_t\boldsymbol{x}_{t-1}$ will first modify $\boldsymbol{g}_t$ (see line 6) and then affect the momentum term $\boldsymbol{m}_t$ (see line 8). While the smoothing of the weight decay factor by $\beta_1$

---

**Algorithm 1** SGD with momentum and SGDW with momentum

1: **given** learning rate $\alpha_t \in \mathbb{R}$, momentum factor $\beta_1 \in \mathbb{R}$, weight decay factor $w \in \mathbb{R}$
2: **initialize** time step $t \leftarrow 0$, parameter vector $\boldsymbol{x}_{t=0} \in \mathbb{R}^n$, first moment vector $\boldsymbol{m}_{t=0} \leftarrow \boldsymbol{0}$, schedule multiplier $\eta_{t=0} \in \mathbb{R}$
3: **repeat**
4:     $t \leftarrow t + 1$
5:     $\nabla f_t(\boldsymbol{x}_{t-1}) \leftarrow SelectBatch(\boldsymbol{x}_{t-1})$     ▷ select batch and return the corresponding gradient
6:     $\boldsymbol{g}_t \leftarrow \nabla f_t(\boldsymbol{x}_{t-1})$ $+w_t\boldsymbol{x}_{t-1}$
7:     $\eta_t \leftarrow SetScheduleMultiplier(t)$     ▷ can be fixed, decay, be used for warm restarts
8:     $\boldsymbol{m}_t \leftarrow \beta_1\boldsymbol{m}_{t-1} + \eta_t\alpha_t\boldsymbol{g}_t$
9:     $\boldsymbol{x}_t \leftarrow \boldsymbol{x}_{t-1} - \boldsymbol{m}_t$ $-\eta_t w_t\boldsymbol{x}_{t-1}$
10: **until** *stopping criterion is met*
11: **return** optimized parameters $\boldsymbol{x}_t$

---

**Algorithm 2** Adam and AdamW

1: **given** $\alpha_t = 0.001, \beta_1 = 0.9, \beta_2 = 0.999, \epsilon = 10^{-8}, w \in \mathbb{R}$
2: **initialize** time step $t \leftarrow 0$, parameter vector $\boldsymbol{x}_{t=0} \in \mathbb{R}^n$, first moment vector $\boldsymbol{m}_{t=0} \leftarrow \boldsymbol{0}$, second moment vector $\boldsymbol{v}_{t=0} \leftarrow \boldsymbol{0}$, schedule multiplier $\eta_{t=0} \in \mathbb{R}$
3: **repeat**
4:     $t \leftarrow t + 1$
5:     $\nabla f_t(\boldsymbol{x}_{t-1}) \leftarrow SelectBatch(\boldsymbol{x}_{t-1})$     ▷ select batch and return the corresponding gradient
6:     $\boldsymbol{g}_t \leftarrow \nabla f_t(\boldsymbol{x}_{t-1})$ $+w_t\boldsymbol{x}_{t-1}$
7:     $\boldsymbol{m}_t \leftarrow \beta_1\boldsymbol{m}_{t-1} + (1 - \beta_1)\boldsymbol{g}_t$     ▷ here and below all operations are element-wise
8:     $\boldsymbol{v}_t \leftarrow \beta_2\boldsymbol{v}_{t-1} + (1 - \beta_2)\boldsymbol{g}_t^2$
9:     $\hat{\boldsymbol{m}}_t \leftarrow \boldsymbol{m}_t/(1 - \beta_1^t)$     ▷ here, $\beta_1$ is taken to the power of $t$
10:     $\hat{\boldsymbol{v}}_t \leftarrow \boldsymbol{v}_t/(1 - \beta_2^t)$     ▷ here, $\beta_2$ is taken to the power of $t$
11:     $\eta_t \leftarrow SetScheduleMultiplier(t)$     ▷ can be fixed, decay, be used for warm restarts
12:     $\boldsymbol{x}_t \leftarrow \boldsymbol{x}_{t-1} - \eta_t\left(\alpha_t\hat{\boldsymbol{m}}_t/(\sqrt{\hat{\boldsymbol{v}}_t} + \epsilon) +w_t\boldsymbol{x}_{t-1}\right)$
13: **until** *stopping criterion is met*
14: **return** optimized parameters $\boldsymbol{x}_t$

---

(see line 8) might be a feature, we note (for simplicity, we omit $\eta_t$) that $\boldsymbol{x}_t$ will decay by $\alpha_t w_t\boldsymbol{x}_{t-1}$ (see line 9) and not $w_t\boldsymbol{x}_{t-1}$ as one could expect according to the definition of the weight decay given by Eq. (1). Practically, if one wants to keep the actual weight decay $\alpha_t w_t$ fixed while changing $\alpha_t$ to $\alpha'_t$, then $w_t$ should be modified to $w'_t = \frac{\alpha_t w_t}{\alpha'_t}$. **This renders the problem of hyperparameter selection of $\alpha_t$ and $w_t$ non-separable**.

We propose to fix this problem by following the original definition of weight decay given by Eq. (1) and decay the weights simultaneously with the update of $\boldsymbol{x}_t$ based on gradient information in Line 9 of Algorithm 1. This yields our proposed SGD variant **SGDW** with momentum. Although the proposed simple modification explicitly decouples $w_t$ and $\alpha_t$, some problem-dependent implicit coupling is likely to remain. In order to account for a possible scheduling of both $\alpha_t$ and $w_t$, we introduce a scaling factor $\eta_t$ delivered by a user-defined procedure $SetScheduleMultiplier(t)$. It should be noted that when $L_2$ regularization is used, weight decay contributes to the batch gradient and thus effectively is scheduled in the same way as the learning rate. Now, since we decouple the two we should also remember to schedule both of them with $\eta_t$.

Having shown that using $L_2$ regularization instead of weight decay already couples regularization and learning rate in the simple case of SGD with momentum, we now consider adaptive gradient optimizers, such as the Adam algorithm proposed by Kingma & Ba (2014), in which the coupling leads to even more unintended behavior. As an adaptive gradient method, Adam maintains a vector $\boldsymbol{v}_t$ responsible for storing smoothed amplitudes of parameter-wise gradients $\boldsymbol{g}_t^2$ (see line 8 in Algorithm 2). These factors are used to control parameter-wise learning rates by normalizing parameter-wise gradients by $\sqrt{\hat{\boldsymbol{v}}_t} + \epsilon$ in line 12 of Algorithm 2. The common way to introduce the weight decay $w_t\boldsymbol{x}_{t-1}$ to Adam results in an update which only distantly resembles the original weight decay given

by Eq. (1) because the $\boldsymbol{v}_t$ vectors are not only responsible for the parameter-wise amplitudes of $\boldsymbol{g}_t$ but also for the parameter-wise amplitudes of weights $\boldsymbol{x}_t$. The amplitudes are then used to re-normalize $\hat{\boldsymbol{m}}_t$ as given in line 12 of Algorithm 2. To gain a bit of intuition, let us consider the case when $t$ is large, causing $\beta_1^t$ and $\beta_2^t$ to go to zero and

$$\boldsymbol{x}_t \leftarrow \boldsymbol{x}_{t-1} - \eta_t \alpha_t \frac{\beta_1 \boldsymbol{m}_{t-1} + (1 - \beta_1)\boldsymbol{g}_t}{\sqrt{\beta_2 \boldsymbol{v}_{t-1} + (1 - \beta_2)\boldsymbol{g}_t^2} + \epsilon}, \text{ with } \boldsymbol{g}_t = \nabla f_t(\boldsymbol{x}_{t-1}) + w_t \boldsymbol{x}_{t-1}, \quad (4)$$

where operations are performed parameter-wise. Not only the batch gradient $\nabla f_t(\boldsymbol{x}_{t-1})$ is normalized but also the weight decay $w_t \boldsymbol{x}_{t-1}$ itself. Since this formula normalizes updates by their typical amplitudes, the decay of weights does not account for amplitudes anymore, leading to the relative decay being weaker for weights with large gradients. **This is a correct implementation of $\mathbf{L}_2$ regularization, but *not* of weight decay.** Therefore, it might be misleading to use the two terms interchangeably, as is commonly done in the literature. We note that this difference between the two mechanisms for Adam has not been investigated and/or described before. As in the case of SGDW, we propose to follow the original definition of weight decay and perform it simultaneously with the gradient-based update as shown in line 12 of Algorithm 2 for **AdamW**. As we will demonstrate experimentally (in Section 5.2), AdamW generalizes much better than Adam.

## 3  NORMALIZED WEIGHT DECAY

Since our preliminary experiments showed that different weight decay factors are optimal for different computational budgets (defined in terms of the number of batch passes), we introduce a normalized weight decay to reduce this dependence. At iteration $t$, $w_t$ is set as follows:

$$w_t = w_{norm} \sqrt{\frac{b_t}{BT_i}}, \quad (5)$$

where $b_t$ is the batch size, $B$ is the total number of training points to be used in one epoch and $T_i$ is the total number of epochs within the $i$-th run/restart of the algorithm. Thus, $w_{norm}$ can be interpreted as the weight decay to be used if only one batch pass is allowed. We note a recent relevant observation of Li et al. (2017) who demonstrated that a smaller batch size (for the same total number of epochs) leads to the shrinking effect of weight decay being more pronounced. Here, we propose to address that effect with normalized weight decay.

## 4  ADAM WITH WARM RESTARTS AND NORMALIZED WEIGHT DECAY

We now apply warm restarts to Adam, following the recent work of Loshchilov & Hutter (2016). There, the authors proposed Stochastic Gradient Descent with Warm Restarts (SGDR) to improve anytime performance of SGD by quickly cooling down the learning rate and periodically increasing it. SGDR has been successfully adopted to lead to new state-of-the-art results for popular image classification benchmarks (Huang et al., 2017; Gastaldi, 2017), and we therefore tried extending it to Adam. However, while our initial version of Adam with warm restarts had better anytime performance than Adam, it was not competitive with SGD with warm restarts, precisely because of Adam's dysfunctional weight decay. Now, having fixed weight decay regularization (Section 2) and also having introduced normalized weight decay (Section 3), the work of Loshchilov & Hutter (2016) on warm restarts directly carries over, and we use it to construct AdamWR to fully benefit from warm restarts.

In the interest of keeping the presentation self-contained, we briefly describe how SGDR schedules the change of the effective learning rate in order to accelerate the training of DNNs. Here, we decouple the initial learning rate and its multiplier $\eta_t$ used to obtain the actual learning rate at iteration $t$ (see, e.g., line 8 in Algorithm 1). In SGDR, we simulate a new warm-started run/restart of SGD once $T_i$ epochs are performed, where $i$ is the index of the run. Importantly, the restarts are not performed from scratch but emulated by increasing $\eta_t$ while the old value of $\boldsymbol{x}_t$ is used as an initial

solution. The amount by which $\eta_t$ is increases controls to which extent the previously acquired information (e.g., momentum) is used. Within the $i$-th run, the value of $\eta_t$ decays according to the cosine annealing (Loshchilov & Hutter, 2016) for each batch as follows:

$$\eta_t = \eta_{min}^{(i)} + 0.5(\eta_{max}^{(i)} - \eta_{min}^{(i)})(1 + \cos(\pi T_{cur}/T_i)), \tag{6}$$

where $\eta_{min}^{(i)}$ and $\eta_{max}^{(i)}$ are ranges for the multiplier and $T_{cur}$ accounts for how many epochs have been performed since the last restart. $T_{cur}$ is updated at each batch iteration $t$ and is thus not constrained to integer values. Adjusting (e.g., decreasing) $\eta_{min}^{(i)}$ and $\eta_{max}^{(i)}$ at every $i$-th restart (see also Smith (2016)) could potentially improve performance, but we do not consider that option in our experiments because it would involve additional hyperparameters. For $\eta_{max}^{(i)} = 1$ and $\eta_{min}^{(i)} = 0$, one can simplify Eq. (6) to

$$\eta_t = 0.5 + 0.5\cos(\pi T_{cur}/T_i). \tag{7}$$

In order to maintain a good anytime performance, one can start with an initially small $T_i$ (e.g., from 1% to 10% of the expected total budget) and multiply it by a factor of $T_{mult}$ (e.g., $T_{mult} = 2$) at every restart. The $(i + 1)$-th restart is triggered when $T_{cur} = T_i$ by setting $T_{cur}$ to 0. An example setting of the schedule multiplier is given in Section 1.1 of the supplementary material. Note that the effective learning rate is controlled by $\eta_t\alpha_t$ where $\alpha_t$ is set to the initial learning rate and stays constant in our experimental setup. The reason why we employ $\alpha_t$ and not simply $\alpha$ is to account for possible practical extensions, e.g., to adapt $\alpha_t$ as a function of batch size in (scheduled) large-batch settings.

Our proposed **AdamWR** algorithm represents AdamW given in Algorithm 2 with $\eta_t$ following Eq. (7) and $w_t$ computed at each iteration using normalized weight decay according to Eq. (5). We note that normalized weight decay allowed us to use a constant parameter setting across short and long runs performed within AdamWR. Equivalently to AdamWR, we define SGDWR as SGDW with warm restarts.

## 5 EXPERIMENTAL VALIDATION

Our experimental setup follows that of Gastaldi (2017), who proposed, in addition to L$_2$ regularization, to apply the new Shake-Shake regularization to a 3-branch residual neural network. Gastaldi (2017) showed that this regularization allowed to achieve new state-of-the-art results of 2.86% on the CIFAR-10 dataset (Krizhevsky, 2009) and of 15.85% on CIFAR-100. The network was trained by SGDR with batch size 128 for 1800 epochs ($T_0 = 1800$) without restarts with the learning rate scheduled by Eq. (6). The regular data augmentation procedure used for the CIFAR datasets was applied. We used the same model/source code based on fb.resnet.torch [1]. The base networks are a 26 2x64d ResNet (i.e. the network has a depth of 26, 2 residual branches and the first residual block has a width of 64) and 26 2x96d ResNet with 11.6M and 25.6M parameters, respectively. For a detailed description of the network and the Shake-Shake method, we refer the interested reader to Gastaldi (2017).

### 5.1 DECOUPLING THE WEIGHT DECAY AND INITIAL LEARNING RATE PARAMETERS

In order to verify our hypothesis about the coupling of the initial learning rate $\alpha_t$ and the weight decay factor $w_t$, we trained a 2x64d ResNet with cosine annealing for 100 epochs with different settings of $\alpha_t$ and $w_t$. Throughout this paper, we scheduled the learning rate with cosine annealing because it leads to better results than a fixed learning rate (see SuppFigure 1 in the supplementary material). Figure 1 compares SGD vs. SGDW (top row) and Adam vs. AdamW (bottom row). For the case of SGD (Figure 1, top left), weight decay is not decoupled from the learning rate (the common way as described in Algorithm 1), and the figure clearly shows that the basin of best hyperparameter settings (depicted by color and top-10 hyperparameter settings by black circles) is not

---

[1]https://github.com/xgastaldi/shake-shake

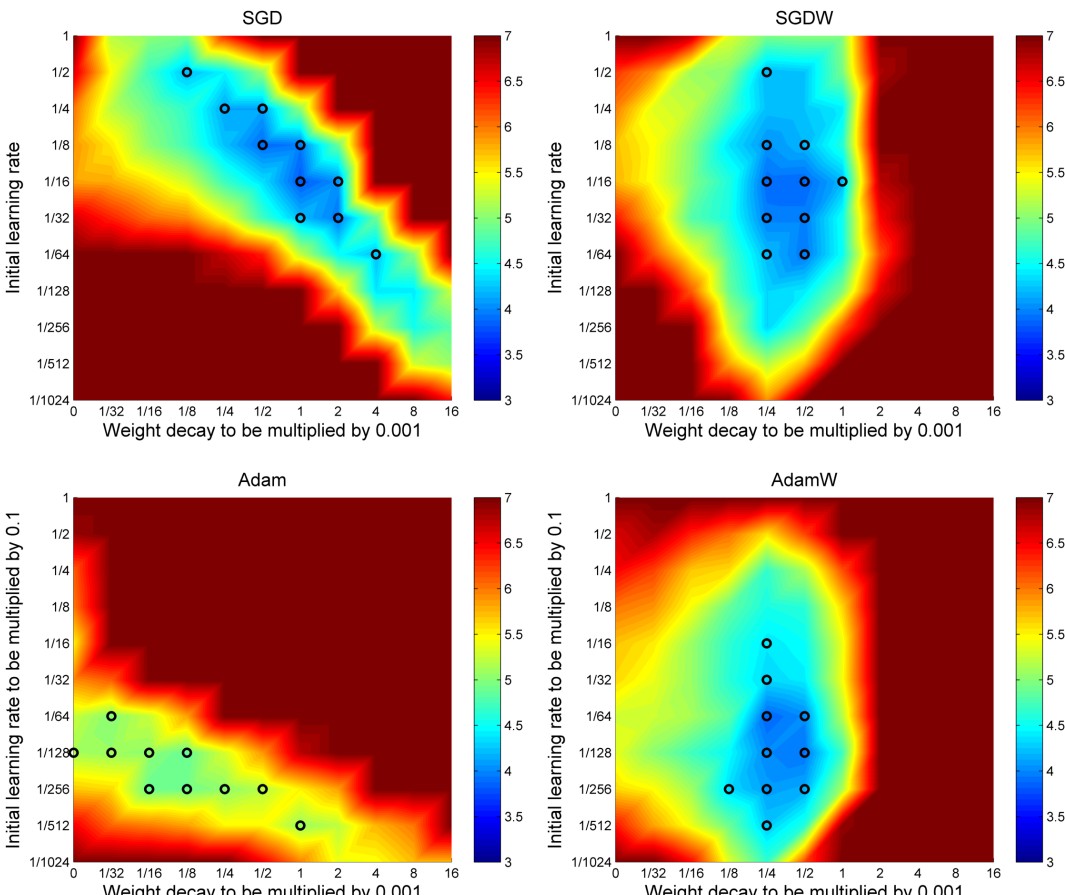

Figure 1: The Top-1 test error of a 26 2x64d ResNet on CIFAR-10 measured after 100 epochs. The proposed SGDW and AdamW (right column) have a more separable hyperparameter space.

aligned with the x-axis or y-axis but lies on the diagonal. This suggests that the two hyperparameters are interdependent and need to be changed simultaneously, while only changing one of them might substantially worsen results. Consider, e.g., the setting at the top left black circle ($\alpha_t = 1/2$, $w_t = 1/8 * 0.001$); only changing either $\alpha_t$ or $w_t$ by itself would worsen results, while changing both of them could still yield clear improvements. We note that this coupling of initial learning rate and weight decay factor might have contributed to SGD's reputation of being very sensitive to its hyperparameter settings.

In contrast, the results for our new SGDW in Figure 1 (top right) show that SGDW decouples weight decay and initial learning rate. The proposed approach renders the two hyperparameters more separable: even if the learning rate is not well tuned yet (e.g., consider the value of 1/1024 in Figure 1, top right), leaving it fixed and only optimizing the weight decay factor would yield a good value (of 1/4*0.001). This is not the case for the original SGD shown in Figure 1 (top left).

The results for different hyperparameter settings of the original Adam are given in Figure 1 (bottom left). Adam's best hyperparameter settings performed clearly worse than SGD's best ones (compare Figure 1, top left). While both methods use the original way to employ weight decay, the original Adam did not benefit from it at all: its best results obtained for non-zero weight decay values were comparable to the best ones obtained without the weight decay regularization, i.e., when $w_t = 0$. Similarly to the original SGD, the shape of the hyperparameter landscape suggests that the two hyperparameters are coupled.

In contrast, the results for our new AdamW in Figure 1 (bottom right) show that AdamW largely decouples weight decay and learning rate. The results for the best hyperparameter settings were substantially better than the best ones of the original Adam and rivaled those of SGD and SGDW.

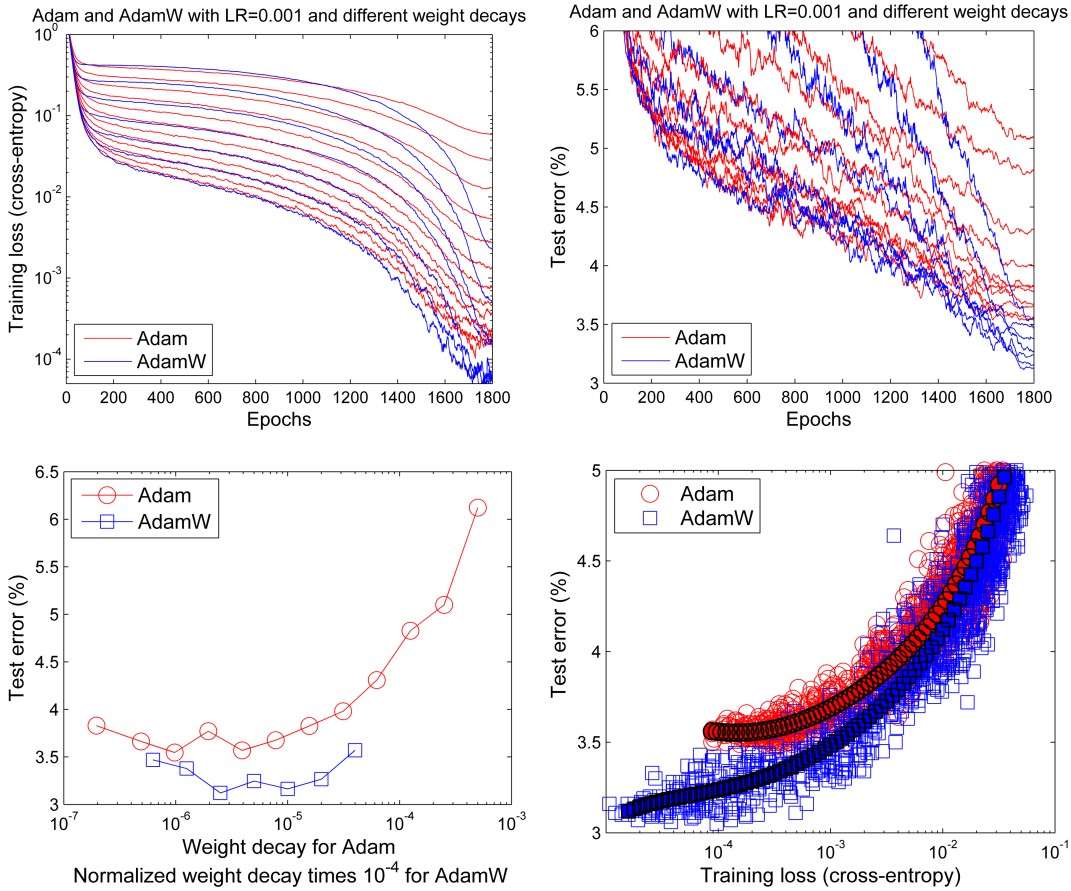

Figure 2: Learning curves (top row) and generalization results (bottom row) obtained by a 26 2x96d ResNet trained with Adam and AdamW on CIFAR-10. See text for details.

In summary, the experimental results in Figure 1 support our hypothesis that the weight decay and learning rate hyperparameters can be decoupled, and that this in turn simplifies the problem of hyperparameter tuning in SGD and improves Adam's performance to be competitive w.r.t. SGD with momentum.

## 5.2 Better Generalization of AdamW

While the previous experiment suggested that the basin of optimal hyperparameters of AdamW is broader and deeper than the one of Adam, we next investigated the results for much longer runs of 1800 epochs to compare the generalization capabilities of AdamW and Adam.

We fixed the initial learning rate to 0.001 which represents both the default learning rate for Adam and the one which showed reasonably good results in our experiments. Figure 2 shows the results for 12 settings of the weight decay of Adam and 7 settings of the normalized weight decay of AdamW. Interestingly, while the dynamics of the learning curves of Adam and AdamW often coincided for the first half of the training run, AdamW often led to lower training loss and test errors (see Figure 2 top left and top right, respectively). Importantly, the use of weight decay in Adam did not yield as good results as in AdamW (see also Figure 2, bottom left). Next, we investigated whether AdamW's better results were only due to better convergence or due to better generalization. **The results in Figure 2 (bottom right) for the best settings of Adam and AdamW suggest that AdamW did not only yield better training loss but also yielded better generalization performance for similar training loss values**. The results on ImageNet32x32 (see SuppFigure 4 in the supplementary material) lead to the same conclusion of substantially improved generalization performance.

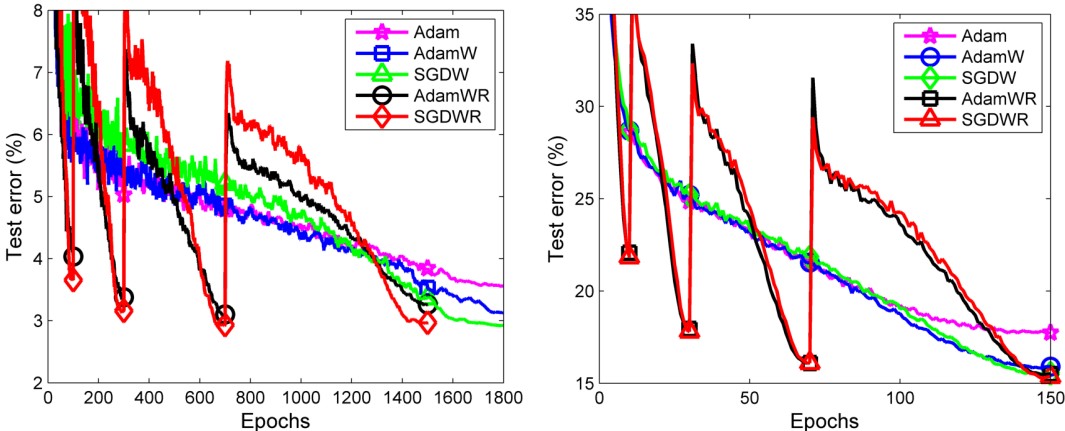

Figure 3: Top-1 test error on CIFAR-10 (left) and Top-5 test error on ImageNet32x32 (right).

## 5.3 Easier Hyperparameter Selection due to Normalized Weight decay

Our experimental results with Adam and SGD suggested that the total runtime in terms of the number of epochs affect the basin of optimal hyperparameters (see SuppFigure 3 in the supplementary material). More specifically, the greater the total number of epochs the smaller the values of the weight decay should be. SuppFigure 3 shows that our remedy for this problem, the normalized weight decay defined in Eq. (7), simplifies hyperparameter selection because the optimal values observed for short runs are similar to the ones for much longer runs. While our initial experiments on CIFAR-10 suggested the square root fit we proposed in Eq. (7), to double-check that this is not a coincidence, we also performed experiments on the ImageNet32x32 dataset (Chrabaszcz et al., 2017), a downsampled version of the original ImageNet dataset with 1.2 million 32×32 pixels images, where an epoch is 24 times longer than on CIFAR-10. This experiment also supported the square root scaling: the best values of the normalized weight decay observed on CIFAR-10 represented nearly optimal values for ImageNet32x32 (see SuppFigure 3). In contrast, had we used the same raw weight decay values $w_t$ for ImageNet32x32 as for CIFAR-10 and for the same number of epochs, **without the proposed normalization, $w_t$ would have been roughly 5 greater than optimal for ImageNet32x32, leading to much worse performance**. The optimal normalized weight decay values were also very similar (e.g., $w_{norm} = 0.025$ and $w_{norm} = 0.05$) across SGDW and AdamW.

We investigated whether the use of much longer runs (1800 epochs) of the original Adam with $L_2$ regularization makes the use of cosine annealing unnecessary. The results of Adam without cosine annealing (i.e., with fixed learning rate) for a 4 by 4 logarithmic grid of hyperparameter settings are given in SuppFigure 5 in the supplementary material. Even after taking into account the low resolution of the grid, the results appear to be at best comparable to the ones obtained with AdamW with 18 times less epochs and a smaller network (see SuppFigure 2). These results are not very surprising given Figure 1 (which demonstrates the effectiveness of AdamW) and SuppFigure 2 (which demonstrates the necessity to use some learning rate schedule such as cosine annealing).

## 5.4 AdamWR with Warm Restarts for better anytime performance

Finally, we investigated the strong anytime performance AdamWR obtains from warm restarts (using normalized weight decay to avoid the need for a different weight decay factor for restarts with longer annealing schedules). As Figure 3 shows, AdamWR greatly sped up AdamW on CIFAR-10 and ImageNet32x32, up to a factor of 10 (see the results at the first restart). For the default learning rate of 0.001, **AdamW achieved 15% relative improvement in test errors compared to Adam both on CIFAR-10** (also see Figure 2) **and ImageNet32x32** (also see SuppFigure 4). **AdamWR achieved the same improved results but with a much better anytime performance.** These improvements closed most of the gap between Adam and SGDWR on CIFAR-10 and yielded comparable performance on ImageNet32x32.

## 6    DISCUSSION AND CONCLUSION

Following suggestions that adaptive gradient methods such as Adam might lead to worse generalization than SGD with momentum (Wilson et al., 2017), we identified at least one possible explanation to this phenomenon: the dysfunctional use of $L_2$ regularization and weight decay. We proposed a simple fix to deal with this issue, yielding substantially better generalization performance in our AdamW variant. We also proposed normalized weight decay and warm restarts for Adam, showing that a more robust hyperparameteer selection and a better anytime performance can be achieved in our new AdamWR variant.

Our preliminary results obtained with AdamW and AdamWR on image classification datasets must be verified on a wider range of tasks, especially the ones where the use of regularization is expected to be important. It would be interesting to integrate our findings on weight decay into other methods which attempt to improve Adam, e.g, normalized direction-preserving Adam (Zhang et al., 2017). While we focussed our experimental analysis on Adam, we believe that similar results also hold for other adaptive gradient methods, such as AdaGrad (Duchi et al., 2011) and RMSProp (Tieleman & Hinton, 2012).

The results shown in Figure 2 suggest that Adam and AdamW follow very similar curves most of the time until the third phase of the run where AdamW starts to branch out to outperform Adam. As pointed out by an anonymous reviewer, it would be interesting to investigate what causes this branching and whether the desired effects are observed at the bottom of the landscape. One could investigate this using the approach of Im et al. (2016) to switch from Adam to AdamW at a given epoch index. Since it is quite possible that the effect of regularization is not that pronounced in the early stages of training, one could think of designing a version of Adam which exploits this by being fast in the early stages and well-regularized in the late stages of training. The latter might be achieved with a custom schedule of the weight decay factor.

In this paper, we argue that the popular interpretation that weight decay = $L_2$ regularization is not precise. Instead, the difference between the two leads to the following important consequences. Two algorithms as different as SGD and Adam will exhibit different effective rates of weight decay even if the same regularization coefficient is used to include $L_2$ regularization in the objective function. Moreover, when decoupled weight decay is applied, two algorithms as different as SGDW and AdamW will optimize two effectively different objective functions even if the same weight decay factor is used. Our findings suggest that the original Adam algorithm with $L_2$ regularization affects effective rates of weight decay in a way that precludes effective regularization, and that effective regularization is achievable by decoupling the weight decay.

Advani & Saxe (2017) analytically showed that in the limited data regime of deep networks the presence of eigenvalues that are zero forms a frozen subspace in which no learning occurs and thus smaller (e.g., zero) initial weight norms should be used to achieve best generalization results. Our future work shall consider adapting initial weight norms or weight norm constraints (Salimans & Kingma, 2016) at each warm restart. Kawaguchi et al. (2017) proposed a family of regularization techniques which are specific to the current batch and its size. Similarly to $L_2$ regularization and weight decay, the latter techniques might be attempted to be transformed to act directly on weights.

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

# 1 SUPPLEMENTARY MATERIAL

## 1.1 AN EXAMPLE SETTING OF THE SCHEDULE MULTIPLIER

An example schedule of the schedule multiplier $\eta_t$ is given in SuppFigure 1 for $T_{i=0} = 100$ and $T_{mult} = 2$. After the initial 100 epochs the learning rate will reach 0 because $\eta_{t=100} = 0$. Then, since $T_{cur} = T_{i=0}$, we restart by resetting $T_{cur} = 0$, causing the multiplier $\eta_t$ to be reset to 1 due to Eq. (7). This multiplier will then decrease again from 1 to 0, but now over the course of 200 epochs because $T_{i=1} = T_{i=0}T_{mult} = 200$. Solutions obtained right before the restarts, when $\eta_t = 0$ (e.g., at epoch indexes 100, 300, 700 and 1500 as shown in SuppFigure 1) are recommended by the optimizer as the solutions, with more recent solutions prioritized.

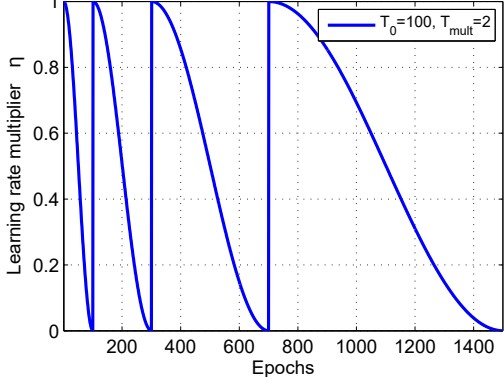

SuppFigure 1: An example schedule of the learning rate multiplier as a function of epoch index. The first run is scheduled to converge at epoch $T_{i=0} = 100$, then the budget for the next run is doubled as $T_{i=1} = T_{i=0}T_{mult} = 200$, etc.

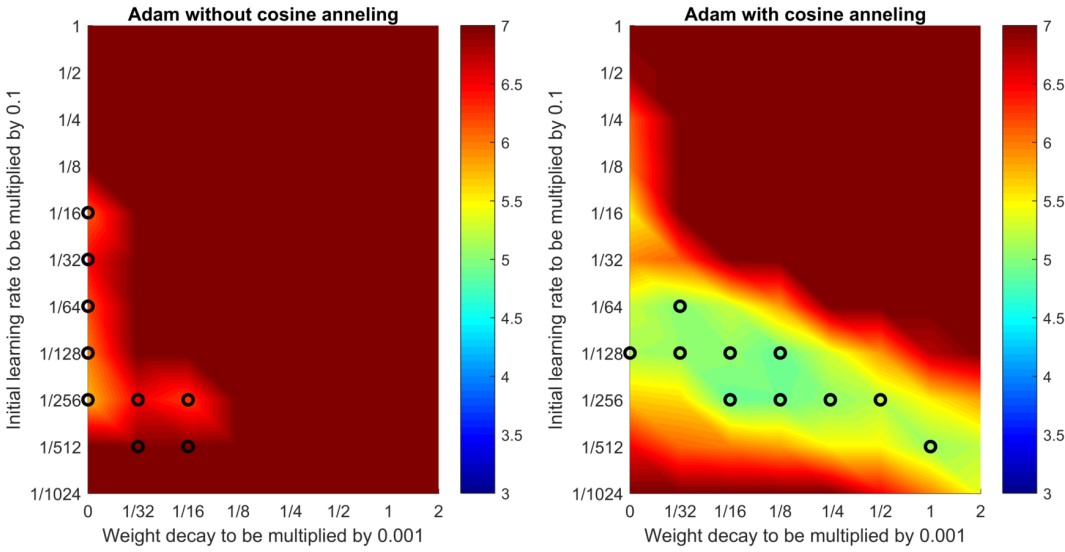

SuppFigure 2: Adam with fixed learning rate (left) and with cosine annealing (right). We show the final test error of a 26 2x64d ResNet on CIFAR-10 after 100 epochs of SGD with momentum. The results where the learning rate is fixed (left) are inferior to the ones where the learning rate is scheduled according to cosine annealing (right). Therefore, we schedule the learning rate with cosine annealing for all methods given in the paper.

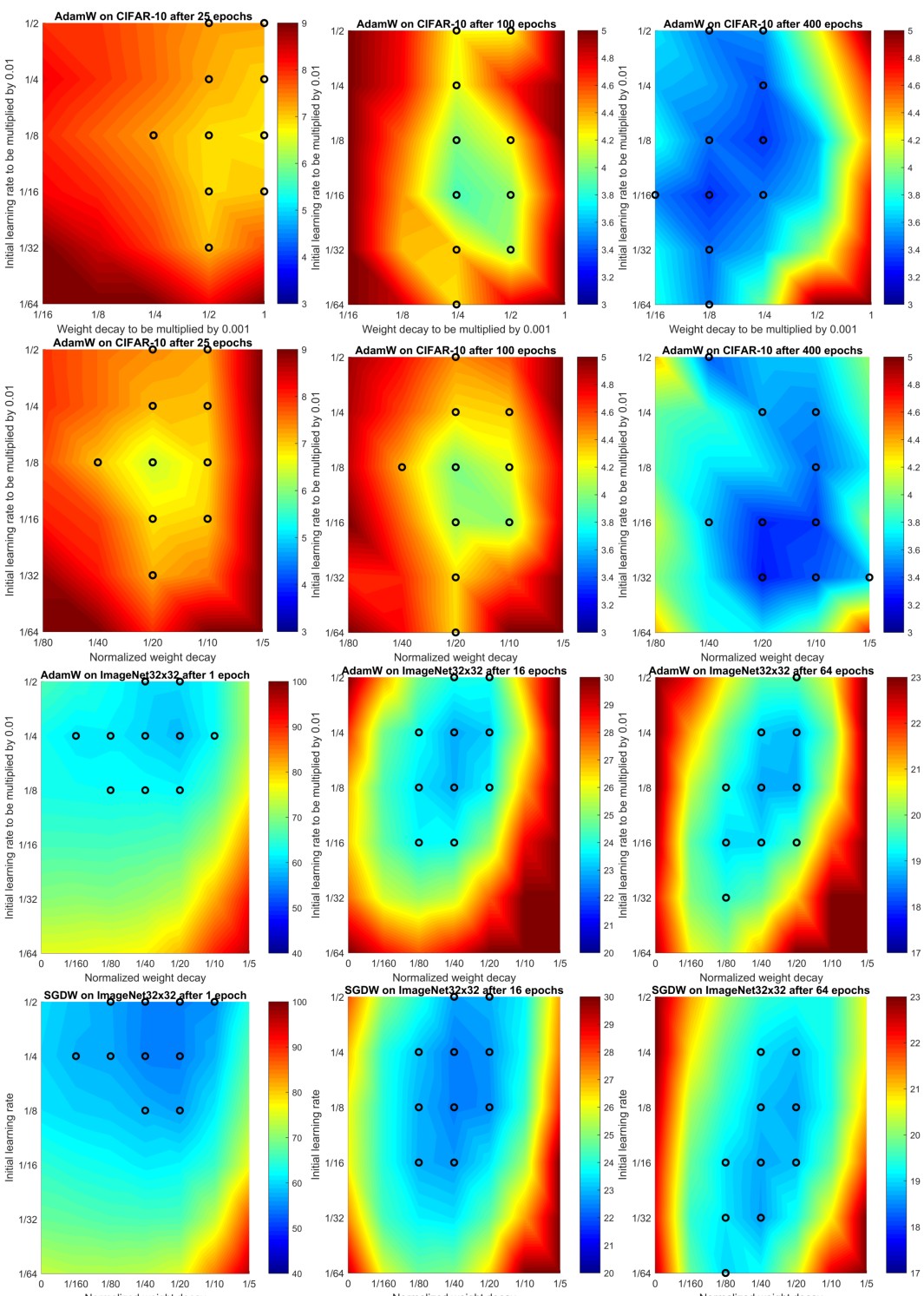

SuppFigure 3: Effect of normalized weight decay. We show the final test Top-1 error on CIFAR-10 (first two rows for AdamW without and with normalized weight decay) and Top-5 error on ImageNet32x32 (last two rows for AdamW and SGDW, both with normalized weight decay) of a 26 2x64d ResNet after different numbers of epochs (see columns). While the optimal settings of the raw weight decay change significantly for different runtime budgets (see the first row), the values of the normalized weight decay remain very similar for different budgets (see the second row) and different datasets (here, CIFAR-10 and ImageNet32x32), and even across AdamW and SGDW.

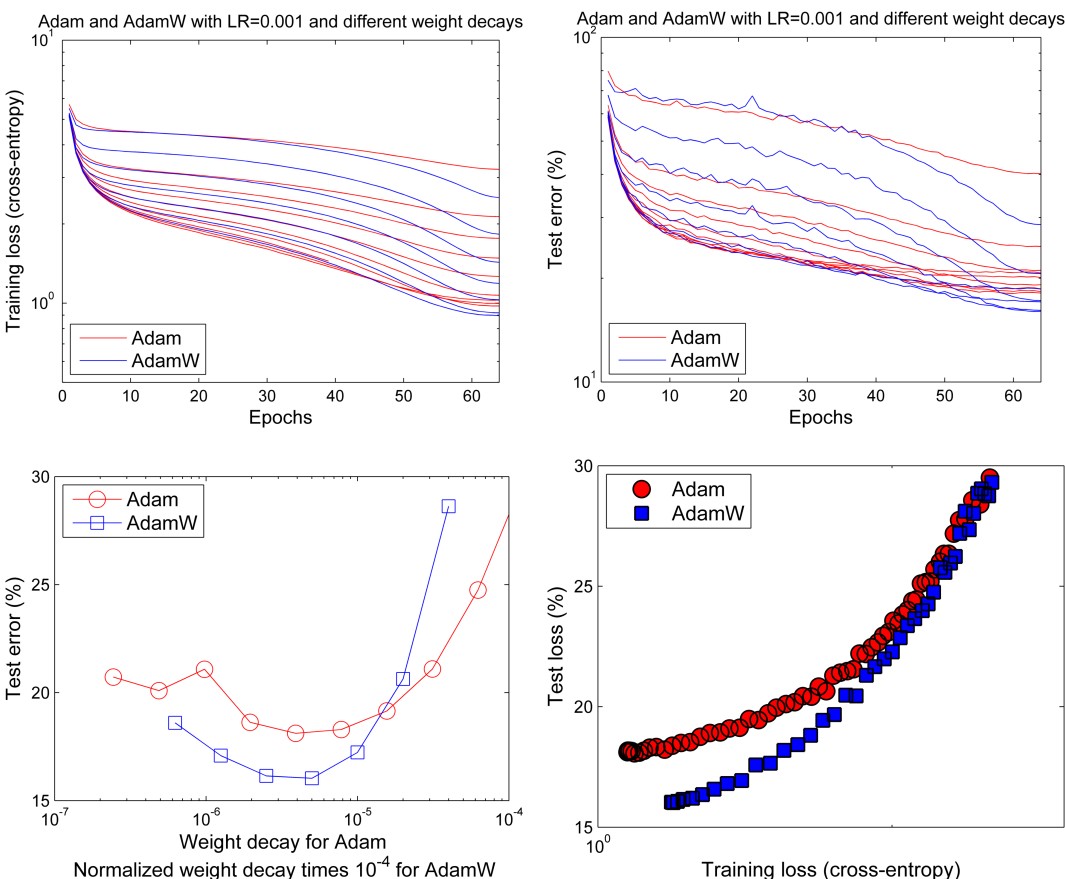

SuppFigure 4: Learning curves (top row) and generalization results (Top-5 errors in bottom row) obtained by a 26 2x96d ResNet trained with Adam and AdamW on ImageNet32x32.

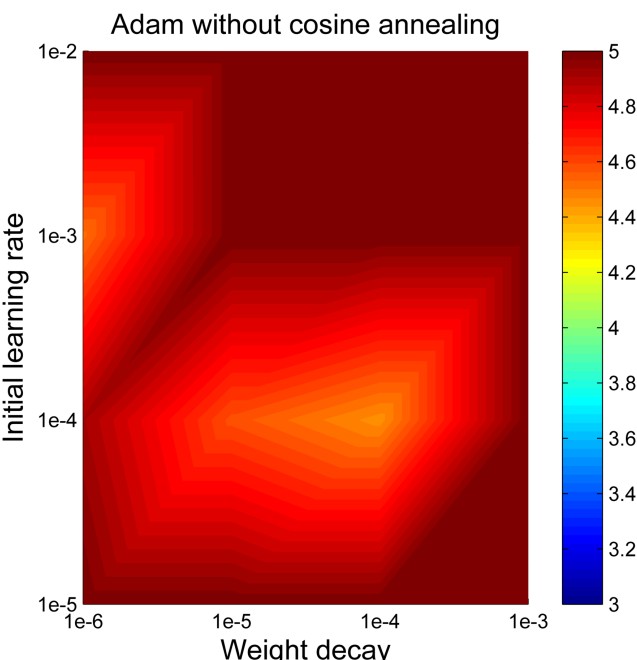

SuppFigure 5: Adam without cosine annealing, i.e., with fixed learning rate. We show the final test error of a 26 2x96d ResNet on CIFAR-10 after 1800 epochs of the original Adam for different settings of learning rate and weight decay used for $L_2$ regularization. These results can be compared to the ones of AdamW shown in SuppFigure 3 (top row). The results of AdamW with only 100 epochs and a smaller network seem to be at least as good as the ones of Adam with 18 times as many epochs and a bigger network.

