# OpenReview forum: "Fixing Weight Decay Regularization in Adam"
_ICLR.cc/2018/Conference — Reject_

### Official Review · AnonReviewer2 · 2017-11-26
**Novel investigation and insight about weight decay in SGD variants**

**Rating:** 8
**Confidence:** 3

**Review:**

This paper investigates weight decay issues lied in the SGD variants, especially Adam. Current implementations of adaptive gradient algorithms implicitly contain a crucial flaw, by which weight decay in these methods does not correspond to L2 regularization. To fix this issue, this paper proposes the decoupling method between weight decay and the gradient-based update.

Overall, this paper is well-written and contain sufficient references to note the overview of recent adaptive gradient-based methods for DNN. In addition, this paper investigates the crucial issue in the recent adaptive gradient methods and find the problem in weight decay. This is an interesting finding. And the proposed method to fix this issue is simple and reasonable. Their experimental results to validate the effectiveness of their proposed method are well-organized. In particular, the investigation on hyperparameter spaces shows the strong advantage of the proposed methods.

---

### Official Review · AnonReviewer3 · 2017-11-26

**Rating:** 4
**Confidence:** 4

**Review:**

The paper presents an alternative way to implement weight decay in Adam. Empirical results are shown to support this idea.

The idea presented in the paper is interesting, but I have some concerns about it.

First, the authors argue that the weight decay should be implemented in a way different from the minimization of a L2 regularization. This seems a very weird statement to me. In fact, it easy to see that what the authors propose is to minimize two different objective functions in SGDW and AdamW! I am not even sure how I should interpret what they propose. The fact is that SGD and Adam are optimization algorithms, so we cannot just change the update rule in the same way in both algorithms and expect them to behave in the same way just because the added terms have the same shape!

Second, the equation (5) that re-normalize the weight decay parameter as been obtained on one dataset, as the author admit, and tested only on another one. I am not sure this is enough to be considered as a scientific proof.

Also, the empirical experiments seem to use the cosine annealing of the learning rate. This means that the only thing the authors proved is that their proposed change yields better results when used with a particular setting of the cosine annealing. What happens in the other cases?

To summarize, I think the idea is interesting but the paper might not be ready to be presented in a scientific conference.

---

> ### Author Response · Authors · 2018-01-05
> **Re: Review**
>
> “the equation (5) that re-normalize the weight decay parameter as been obtained on one dataset, as the author admit, and tested only on another one.”
> While we don’t have evidence that the sqrt scaling we propose is optimal, we believe that *some* scaling should be considered when the total number of batch passes changes (due to the change of the total number of epochs or/and batch size). It is not (computationally) straightforward to investigate the optimal scaling because it is coupled with other hyperparameters. We note, however, that our focused study on CIFAR-10 and ImageNet32x32 represents the first attempt in this direction, and that it at least demonstrates that in these two cases, sqrt scaling is much better than the previous default (no scaling).
>
> “Also, the empirical experiments seem to use the cosine annealing of the learning rate. This means that the only thing the authors proved is that their proposed change yields better results when used with a particular setting of the cosine annealing. What happens in the other cases?”
>
> We note that we experimented with and presented a set of results/figures with different settings of cosine annealing (varying its initial learning rate). As discussed in Section 2, the separability effect provided by the proposed decoupling does not rely on cosine annealing. In response to the reviewer’s comment, we have now also included SuppFigure 5 (for the moment, unfortunately, it is not of the greatest possible resolution due to its high computational cost) which shows the results without cosine annealing. We included the following text in section 5.3.
>
> "We investigated whether the use of much longer runs (1800 epochs) of the original Adam with L2 regularization makes the use of cosine annealing unnecessary. The results of Adam without cosine annealing (i.e., with fixed learning rate) for a 4 by 4 logarithmic grid of hyperparameter settings are given in SuppFigure 5 in the supplementary material. Even after taking into account the low resolution of the grid, the results appear to be at best comparable to the ones obtained with AdamW with 18 times less epochs and a smaller network (see SuppFigure 2). These results are not very surprising given Figure 1 (which demonstrates the effectiveness of AdamW) and SuppFigure 2 (which demonstrates the necessity to use some learning rate schedule such as cosine annealing)."
>
>
> We agree that the impact of weight decay on the objective function should be mentioned. We included the following text in our discussion section.
>
> "In this paper, we argue that the popular interpretation that weight decay = L2 regularization is not precise. Instead, the difference between the two leads to the following important consequences. Two algorithms as different as SGD and Adam will exhibit different effective rates of weight decay even if the same regularization coefficient is used to include L2 regularization in the objective function. Moreover, two algorithms as different as SGDW and AdamW will optimize two effectively different objective functions even if the same weight decay factor is used. Our findings suggest that the original Adam algorithm with L2 regularization affects effective rates of weight decay in a way that precludes effective regularization, and that effective regularization is achievable by decoupling the weight decay."

---

### Official Review · AnonReviewer1 · 2017-11-27
**Important work supported with experiments**

**Rating:** 7
**Confidence:** 4

**Review:**

At the heart of the paper, there is a single idea: to decouple the weight decay from the number of steps taken by the optimization process (the paragraph at the end of page 2 is the key to the paper). This is an important and largely overlooked area of implementation and most off-the-shelf optimization algorithms, unfortunately, miss this point, too. I think that the proposed implementation should be taken seriously, especially in conjunction with the discussion that has been carried out with the work of Wilson et al., 2017 (https://arxiv.org/abs/1705.08292).

The introduction does a decent job explaining why it is necessary to pay attention to the norm of the weights as the training progresses within its scope. However, I would like to add a couple more points to the discussion:
- "Optimal weight decay is a function (among other things) of the total number of epochs / batch passes." in principle, it is a function of weight updates. Clearly, it depends on the way the decay process is scheduled. However, there is a bad habit in DL where time is scaled by the number of epochs rather than the number of weight updates which sometimes lead to misleading plots (for instance, when comparing two algorithms with different batch sizes).
- Another ICLR 2018 submission has an interesting take on the norm of the weights and the algorithm (https://openreview.net/forum?id=HkmaTz-0W&noteId=HkmaTz-0W). Figure 3 shows the histograms of SGD/ADAM with and without WD (the *un-fixed* version), and it clearly shows how the landscape appear misleadingly different when one doesn't pay attention to the weight distribution in visualizations.
- In figure 2, it appears that the training process has three phases, an initial decay, a steady progress, and a final decay that is more pronounced in AdamW. This final decay also correlates with the better test error of the proposed method. This third part also seems to correspond to the difference between Adam and AdamW through the way they branch out after following similar curves. One wonders what causes this branching and whether the key the desired effects are observed at the bottom of the landscape.
- The paper concludes with "Advani & Saxe (2017) analytically showed that in the limited data regime of deep networks the presence of eigenvalues that are zero forms a frozen subspace in which no learning occurs and thus smaller (e.g., zero) initial weight norms should be used to achieve best generalization results." Related to this there is another ICLR 2018 submission (https://openreview.net/forum?id=rJrTwxbCb), figure 1 shows that the eigenvalues of the Hessian of the loss have zero forms at the bottom of the landscape, not at the beginning. Back to the previous point, maybe that discussion should focus on the second and third phases of the training, not the beginning.
- Finally, it would also be interesting to discuss the relation of the behavior of the weights at the last parts of the training and its connection to pruning.

I'm aware that one can easily go beyond the scope of the paper by adding more material. Therefore, it is not completely reasonable to expect all such possible discussions to take place at once. The paper as it stands is reasonably self-contained and to the point. Just a minor last point that is irrelevant to the content of the work: The slash punctuation mark that is used to indicate 'or' should be used without spaces as in 'epochs/batch'.

Edit: Thanks very much for the updates and refinements. I stand by my original score and would like to indicate my support for this style of empirical work in scientific conferences.

---

> ### Author Response · Authors · 2018-01-05
> **Re: Important work supported with experiments**
>
> We agree that "number of epochs/batch passes" should be changed to "number of batch passes/weight updates" and fixed this (see Section 1). We also included the following text in Section 3:
>
> "We note a recent relevant observation of \cite{li2017visualizing} who demonstrated that a smaller batch size (for the same total number of epochs) leads to the shrinking effect of weight decay being more pronounced. Here, we propose to address that effect with normalized weight decay."
>
> Following the insight that you provided, we included the following text in our discussion section.
>
> "The results shown in Figure 2 suggest that Adam and AdamW follow very similar curves most of the time until the third phase of the run where AdamW starts to branch out to outperform Adam. As pointed out by an anonymous reviewer, it would be interesting to investigate what causes this branching and whether the desired effects are observed at the bottom of the landscape. One could investigate this using the approach of \cite{im2016empirical} to switch from Adam to AdamW at a given epoch index. Since it is quite possible that the effect of regularization is not that pronounced in the early stages of training, one could think of designing a version of Adam which exploits this by being fast in the early stages and well-regularized in the late stages of training. The latter might be achieved with a custom schedule of the weight decay factor."

---

### Public Comment · (anonymous) · 2017-11-27
**Need to extend domain of plot in Figure 1?**

In Figure 1 top, the blue region of SGDW is fully visible in the plot. But for SGD, the blue region gets chopped off the edge of the plot. This seems to make a fair comparison difficult. In particular, the following statement seems questionable, since it is not clear what happens for SGD outside of the visible region in the plot.

"even if the learning rate is not well tuned yet (e.g., consider the value of 1/1024 in Figure 1, top right), leaving it fixed and only optimizing the weight decay factor would yield a good value (of 1/4*0.001). This is not the case for the original SGD shown in Figure 1 (top left)."

---

> ### Author Response · Authors · 2017-11-28
> **+ corrected sentence, + extended Figure 1**
>
> Thanks for the note!
>
> Even if the shape of the hyperparameter space would drastically change outside of the current range, the claim would be correct for SGD because the already presented results alone make it impossible to first fix the learning rate LR to any value from the range and then expect that the best weight decay found for that LR value would be nearly-optimal for all other possible values of LR. However, we agree that the example given in the sentence is unfortunate because it asks the reader to extrapolate instead of dealing with the data that is shown. It is confusing and we will correct that with a better example whose results are shown in Figure 1:  when LR=0.5, optimal weight decay factor is 1/8 *0.001 but it is not optimal for all other settings of LR.
>
> Regarding the values outside of the current range, it seems very unlikely that better results for LR>0.2 exist given the isolines shown in Figure 1 (note the elliptic shape and that the top results for LR=0.2 are worse than for LR=0.1) and that none of the papers with ResNets on CIFAR-10 (with standard settings of batch size, etc.) we are aware of use LR>0.2. In fact, since momentum-SGD is a standard baseline, its hyperparameters for ResNets on CIFAR-10 have been heavily tuned by researchers so that LR often lies in [0.05, 0.1] that matches the best region of momentum-SGD shown in Figure 1.
>
> Thank you for helping to avoid possible confusions: we will correct the sentence and extend Figure 1 of momentum-SGD by an additional column with LR=0.4 and even larger LR if necessary.

---

> > ### Author Response · Authors · 2018-01-05
> > **Re: + corrected sentence, + extended Figure 1**
> >
> > Following your suggestion, we extended Figure 1 to show the results for much larger weight decay factors. The results confirmed our expectations that the original figures included the basin of optimal hyperparameter settings of the considered experimental setup. You rightly pointed out that a sentence describing Figure 1 was confusing; we have fixed the sentence to provide a more illustrative example.

---

### Public Comment · (anonymous) · 2018-01-04
**Comparisons with SGD and SGDR needed**

This is a very interesting paper and I think optising weight decay is an important under-explored area.

However, I am left in doubt as to the value of the contribution, possibly only because some additional clarity is needed.

For practical purposes, I would like to know whether its worth attempting to use SGDW or SGDWR rather than standard SGD. Its not clear from Figure 3 if it is worth the effort, because there is no direct comparison between your method and standard SGD methods, yet there is a comparison with standard ADAM. Why the omission?

I also note that Figure 3 suggests that Adam variants seems always inferior to comparison vanilla SGD methods, which also leads to the question of why bother "fixing" Adam if SGD variants are better and simpler choices?

It would also be good to know if the W methods work well for more common network variants like standard residual networks.

Finally, due to the efforts to make the algorithms as general as possible, I was left in some confusion about the precise choices of parameters used in the experiments. For example, for the SGDWR results, was w_t constant between restarts and set according to equation (5), and if so what w_norm was used?  In this case, what value of \alpha_t was used?

---

> ### Author Response · Authors · 2018-01-05
> **Re: Comparisons with SGD and SGDR needed**
>
> “For practical purposes, I would like to know whether its worth attempting to use SGDW or SGDWR rather than standard SGD.”
>
> SGDW is worth using if you consider that the search space of hyperparameters of SGDW shown in Figure 1 is easier to search that the one of SGD shown in the same figure. We consider this to be the case due to the more separable nature of that space as described in the paper. Another reason to prefer SGDW to SGD is the proposed normalized weight decay that allows you to simplify the search for the weight decay factor suitable to different computational budgets. Please compare the first two rows of SuppFigure 3: the normalized weight decay factor of 1/20 is suitable for 25 and 400 epochs, in contrast to the raw weight decay factor whose optimal value changes by a factor of about 4.
> As you can see in Figure 1, despite the fact that it is easier to tune SGDW than SGD, the best validation errors that can be obtained by both algorithms are comparable. Therefore, we only claim that SGDW “simplifies the problem of hyperparameter tuning in SGD” and did not run SGD for Figure 3 which would match the results of SGDW (similarly to Figure 1), i.e., reproduce 2.86% of Gastaldi. However, due to a request made to us earlier, we have included an additional line for the ImageNet32x32 experiment (see Figure 3 right): results for original Adam (with cosine annealing). Similarly to the results on CIFAR-10 (see Figure 3 left), the best results of Adam (out of a set of weight decay factors) were substantially worse than the ones of AdamW.
>
> “I also note that Figure 3 suggests that Adam variants seems always inferior to comparison vanilla SGD methods, which also leads to the question of why bother "fixing" Adam if SGD variants are better and simpler choices?”
>
> Please note that Figure 3 shows that the proposed “fixed” Adam drastically reduces the gap between SGD on CIFAR-10 and performs equally well (no longer inferior) on ImageNet32x32. As mentioned at the end of our introduction, our motivation was to contribute to the goal that “practitioners do not need to switch between Adam and SGD anymore, which in turn should help to reduce the common issue of selecting dataset/task-specific training algorithms and their hyperparameters”. “Fixing” Adam for the considered image classification datasets where its gap with SGD is significant might be a good indication of progress towards achieving the above-mentioned goal.
>
> “was w_t constant between restarts and set according to equation (5), and if so what w_norm was used? In this case, what value of \alpha_t was used?”
> w_norm is the normalized weight decay hyperparameter, set to the value indicated in the plots (e.g., as 0.025). It is used to derive w_t according to equation (5). Since all inputs of equation (5) are constant between restarts in our setup, w_t is constant as well. Please note that if batch size would change during the run (e.g., increase), then w_t would change as well.
>
> alpha_t is constant and corresponds to the initial learning rate, then it is multiplied by the schedule multiplier eta_t which includes cosine annealing and restarts

---

### Author Response · Authors · 2018-01-05
**Rebuttal**

We thank all reviewers for their positive evaluation and their valuable comments. We've uploaded a revision to address the issues raised and replied to reviewers and anonymous comments individually in the OpenReview forum.
We are glad that the reviewers agree that our work is novel, simple and might provide useful insights. We agree that some of our experimental findings need to be explored on a wider range of datasets and tasks. Nevertheless, we hope that our paper provides a useful bit of information to better understand regularization of deep neural networks.

Thank you again for your reviews!

---

### Decision · Program_Chairs · 2018-01-29
**ICLR 2018 Conference Acceptance Decision**

**Decision:**

Reject

**Comment:**

This paper generated quite a bit of controversy among reviewers. The main claim of the paper is that Adam and related optimizers are broken because their "weight decay" regularization is not actually weight decay. It proposes to modify Adam to decay all weights the same regardless of the gradient variances.

Calling Adam's weight decay mechanism a mistake seems very far-fetched to me. Neural net optimization researchers are well aware of the connection between weight decay and L2 regularization and the fact that they don't correspond in preconditioned methods. L2 regularization is basically the only justification I have heard for weight decay, and despite rejecting this interpretation, the paper does not provide an alternative justification.

Decoupling the optimization from the cost function is a well-established principle. This abstraction barrier is not completely clean (e.g. gradient noise has well-known regularization effects), and the experiments of this paper perhaps provide evidence that the choices may be coupled in this case. This is an interesting finding, and probably worth following up on. However, the paper seems to sweep the "decoupling optimization and cost" issue under the carpet and take for granted that the decay rate is what should be held fixed. All three reviewers found the presentation to be misleading, and I would agree with them. While there may be an interesting contribution here, I cannot endorse the paper as-is.

---

> ### Author Response · Authors · 2018-02-16
> **Re: ICLR 2018 Conference Acceptance Decision**
>
> We are disappointed about this decision, especially seeing that our average score (6.33) lies in the acceptance region.
> Nevertheless, we have taken the feedback seriously and improved the paper substantially in the meantime; see https://arxiv.org/pdf/1711.05101.pdf
>
> Just for the record (e.g., for any potential future reviewers), we would like to clear up a few points:
>
> > Calling Adam's weight decay mechanism a mistake seems very far-fetched to me.
>
> We agree that this would be far-fetched. Please note that we never said that this is a mistake in Adam; we said that the common *implementation of L2 regularization/weight decay* does not correspond to the original proposal of weight decay regularization for the case of adaptive gradient algorithms, such as Adam. This is not a criticism of Adam, but of the way weight decay regularization is implemented in common deep learning libraries.
>
> > Neural net optimization researchers are well aware of the connection between weight decay and L2 regularization and the fact that they don't correspond in preconditioned methods.
>
> This statement does not correspond with our experience. 10/10 researchers we've asked (from several universities) did not know about this. We note that after hearing about a simple-to-prove fact it is tempting to convince oneself that one always knew this fact. If there exists a prior reference pointing out the inequivalence then we would love to hear about it; but if no such a reference exists we stand by our claims.
>
> > All three reviewers found the presentation to be misleading, and I would agree with them.
>
> We refer to the original reviews below: two of the reviewers were very positive, with scores of 7 and 8.
>
> > L2 regularization is basically the only justification I have heard for weight decay, and despite rejecting this interpretation, the paper does not provide an alternative justification. Decoupling the optimization from the cost function is a well-established principle. This abstraction barrier is not completely clean (e.g. gradient noise has well-known regularization effects), and the experiments of this paper perhaps provide evidence that the choices may be coupled in this case. This is an interesting finding, and probably worth following up on. However, the paper seems to sweep the "decoupling optimization and cost" issue under the carpet and take for granted that the decay rate is what should be held fixed.
>
> We agree that our ICLR submission did not study this in enough detail, and this is the key part of the paper we have improved in the meantime.
> Specifically, in our new version (available at https://arxiv.org/pdf/1711.05101.pdf ), we've added a new Section 3 to formally contrast L2 regularization and weight decay, and to provide intuition about what the latter means for adaptive gradient algorithms. In particular, to relate weight decay to which objective function is being optimized, for the special case of a fixed diagonal preconditioner matrix M = [diag(\vec{s})^{-1}], we derived an equivalence of weight decay to a new version of L2 regularization that takes the norm of weights scaled by the preconditioner. From the angle of which objective function is being optimized, it is thus possible to understand weight decay as follows:
> - if all gradients are typically of equal size, weight decay is the same as L2 regularization
> - if some weights typically have larger gradients they get penalized more heavily (proportionally to their typical gradients).
> This encodes a preference in the objective function not only for solutions with small weights, but particularly with small weights for those weights that tend to have larger gradients than others. One interpretation of why this may work well is that it may drive the search towards flatter minima that generalize better than sharper minima.